# INFOSCAN: INFORMATION-EFFICIENT VISUAL SCANNING VIA RESOURCE-ADAPTIVE WALKS

**Yifeng Wu**[1,2]*, **Huimin Huang**[3]*, **Shangjie Zhou**[4]*, **Yawen Huang**[3], **Hao Zheng**[3],
**Yun Chen**[5], **Xian Wu**[3], **Ruize Han**[1]†, **Guanhua Chen**[2]†
[1]Shenzhen University of Advanced Technology, [2]Southern University of Science and Technology
[3]Tencent Jarvis Lab, [4]Xi'an Jiaotong University, [5]Shanghai University of Finance and Economics

## ABSTRACT

High-resolution visual representation learning remains challenging due to the quadratic complexity of Vision Transformers and the limitations of existing efficient approaches, where fixed scanning patterns in recent Mamba-based models hinder content-adaptive perception. To address these limitations, a novel Information-aware Scanning mechanism (InfoScan) tailored for state-space visual backbones is proposed, which dynamically allocates computational resources to the most salient regions of an image. Specifically, InfoScan rigorously assesses the informativeness of image patches by integrating entropy with local structural analyses, formulates a joint optimization objective balancing fine-grained detail preservation and broader contextual coherence, and learns an adaptive scanning policy via reinforcement learning. Built upon the innovative Visual Information State Space (VISS) block, InfoScan establishes a new family of models that achieve superior efficiency-accuracy trade-offs across diverse tasks. Extensive empirical evaluation in different downstream vision tasks demonstrates that our information-driven dynamic scanning paradigm offers a robust and principled alternative to fixed or global-first traversal methods. Collectively, our work positions adaptive, content-aware processing as a promising and effective new paradigm for efficient high-resolution visual representation. Our code and models are publicly available at https://github.com/SIAT-CV-wuyifeng/Infoscan-ICLR2026.

## 1 INTRODUCTION

Visual representation learning, a cornerstone of computer vision, aims to extract complex patterns from visual data. Vision Transformers (ViTs) (Dosovitskiy et al., 2021; Vaswani et al., 2017) have become a dominant backbone for visual representation learning, achieving widespread success across diverse downstream tasks such as classification, segmentation, and detection. By incorporating self-attention mechanisms, ViTs demonstrate superior learning capacity on large-scale datasets. However, their computational cost scales quadratically with the number of input tokens, making them prohibitively expensive when processing high-resolution images–where the number of tokens grows significantly.

To mitigate this issue, extensive research has focused on reducing computational complexity while preserving performance (Dong et al., 2022; Liu et al., 2021). These approaches typically operate through either token sparsification or hierarchical downsampling. Nevertheless, they still face a fundamental trade-off: methods that restrict token interaction often limit the effective receptive field, while aggressive downsampling leads to non-negligible performance degradation across diverse tasks. Consequently, achieving both efficiency and strong representational power remains an open challenge.

Recently, Mamba-based models such as VMamba (Liu et al., 2024), RainMamba (Wu et al., 2024), and others (Mehta et al., 2023; Zubić et al., 2024; Zhu et al., 2024b; Gu & Dao, 2024) have

---

*Equal Contribution.
†Corresponding authors.

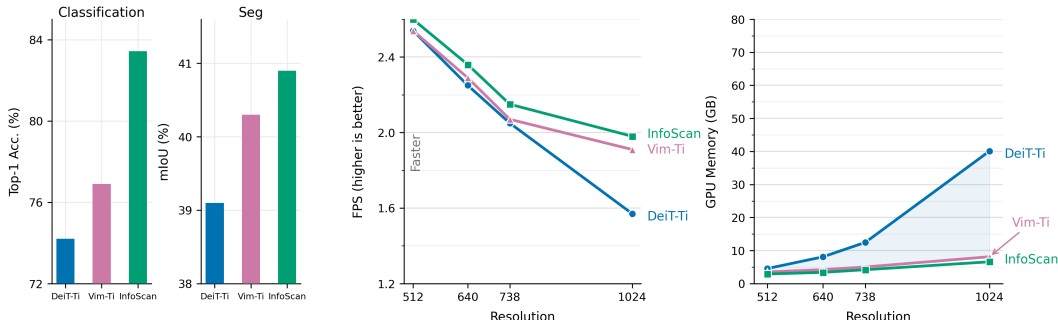

Figure 1: Performance comparison across resolutions for classification accuracy, segmentation mIoU, FPS, and GPU memory usage of DeiT-Ti, Vim-Ti, and InfoScan.

emerged for efficient visual representation learning, leveraging structured scanning patterns within state-space architectures to compress hidden states and capture long-range dependencies. These approaches, built upon predefined traversal orders, have alleviated computational bottlenecks to some extent. However, they perform uniform scanning–such as raster or Hilbert curves–treating all patches identically regardless of their informational content. This rigid scanning paradigm implicitly assumes uniform information distribution across the image, thereby overlooking the varying significance of local regions and limiting adaptive feature aggregation.

In this paper, we propose InfoScan, an information-gain-driven novel Vision model that adaptively allocates computation based on feature significance. Our framework is built upon the VISS (Vision Information State Space) block and consists of three key components: an information scoring module to estimate the informativeness of each patch, a patch selection mechanism to prioritize content-rich regions, and a sequential scanning policy learned via reinforcement to dynamically adjust the processing order. By focusing computation on high-value regions early in the forward pass, InfoScan achieves strong performance with significantly reduced computational overhead, enabling efficient and adaptive vision modeling.

Compared with benchmark vision models based on CNNs (ConvNeXt (Liu et al., 2022a)), ViTs (Swin (Liu et al., 2021), DeiT (Touvron et al., 2021b)), state-space models (Vim (Zhu et al., 2024a)), and SS2D architectures (VMamba (Liu et al., 2024)), InfoScan achieves consistent improvements of $+0.8\%$ to $+1.9\%$ in mIoU and $+0.6\%$ to $+5.8\%$ in Top-1 accuracy across image classification, segmentation, and object detection tasks, while reducing model parameters by 18% to 32%. Notably, on image classification, InfoScan shows particularly strong gains, outperforming all baselines by over $+1.5\%$ Top-1 on ImageNet-1K. Under the Mask R-CNN framework (Han et al., 2021), it outperforms Swin-B and ConvNeXt-B on MSCOCO2017 (Lin et al., 2014) with 30M fewer parameters. These gains are consistent across model scales and domains, including natural and medical imaging.

Our contributions are summarized as follows: (1) We introduce an information-aware scanning mechanism that quantifies the significance of each image patch through a weighted combination of Shannon entropy and local variance, enabling the model to adaptively prioritize high-information regions. (2) We propose a principled mathematical framework to jointly optimize patch information content, information loss, and scanning step size, yielding a more efficient and effective traversal strategy beyond fixed or heuristic scanning paths. (3) We design a reward-driven dynamic scanning policy based on a Markov decision process, allowing the model to learn where to attend next according to contextual information density, thereby enhancing both local detail preservation and global context integration.

## 2 RELATED WORK

### 2.1 EFFICIENT AND ADAPTIVE COMPUTATION IN VISION

Modern vision models face growing computational demands, especially when processing high-resolution inputs. A dominant paradigm for efficiency is sparse computation, which reduces FLOPs

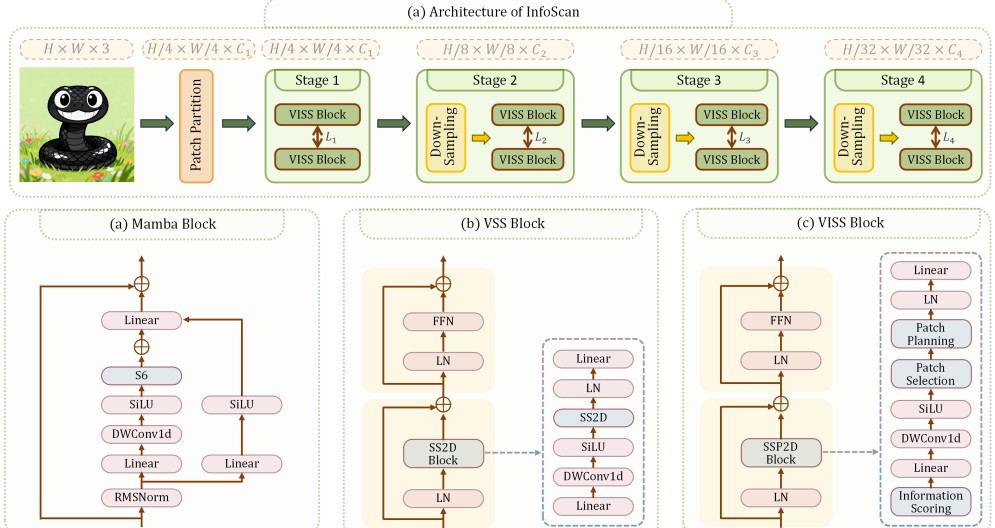

Figure 2: Architecture of InfoScan and its key components. (a) Overall network with patch embedding and hierarchical VISS blocks. Mamba block with RMSNorm, DWConv1d, and linear layers. (b) VSS block integrating SS2D and FFN. (c) VISS block with SPS2D for adaptive scanning.

by selectively activating model components or processing only a subset of visual tokens. Sparse attention mechanisms (Child et al., 2019) restrict contextual interactions to local neighborhoods or top-$k$ salient regions, while dynamic token pruning (Kim et al., 2024) removes low-importance patches during inference. Concurrently, conditional computation approaches (Bengio et al., 2013; Wang et al., 2022) adapt model capacity based on input complexity, such as allocating more resources to semantically rich regions.

However, most existing methods operate under a static or reactive paradigm: they either apply fixed sparsity patterns or reweight tokens after a full forward pass over all patches. In contrast, our work proposes a proactive efficiency strategy–by learning to scan patches in an order that prioritizes high-information regions early, we reduce redundant computation at the input level, before feature aggregation begins. This shifts the efficiency bottleneck from post-hoc pruning to front-loaded perceptual prioritization, aligning with cognitive principles of selective attention (Itti et al., 2001) while maintaining end-to-end trainability.

## 2.2 Scanning Strage

The design of visual scanning policies has long been a foundational consideration in both classical and modern vision architectures. Early approaches, such as raster (Gu & Dao, 2024) and zigzag scanning (Ma et al., 2019), enforce fixed, content-agnostic orders based solely on spatial coordinates. These deterministic strategies are computationally efficient and easy to implement, making them prevalent in standard State Space Models. Alternatively, space-filling curves like Hilbert (Kamata et al., 1999) and Z-order curves aim to enhance spatial coherence by minimizing the Euclidean distance between consecutive patches, improving locality in sequential processing. Despite their geometric elegance, all such methods assume uniform information density across the image—a strong prior that rarely holds in real-world data. This leads to inefficient computation, as high-entropy, semantically rich regions (e.g., object boundaries) are processed no earlier than homogeneous backgrounds. More recent works explore dynamic attention mechanisms to reweight patch importance (Dai et al., 2021), but they still rely on globally scanning all patches first.

Figure 3: A comparison of InfoScan scanning paths with other scanning patterns. The numbers in the figure indicate the number of InfoScan scans.

## 3 METHODOLOGY

Visual scanning defines a policy $\pi$ that maps a 2D grid of image patches $\mathcal{P} = \{p_{i,j}\}$ to a 1D sequence $S_\pi = (s_1, \ldots, s_N)$, where $s_t \in \mathcal{P}$. This sequencing is critical for directional models such as state-space models (SSMs) and causal Transformers, where the processing order directly influences contextual integration and computational efficiency. The core problem is thus to find an optimal policy $\pi^* \in \Pi$ that maximizes a task-specific objective $\mathcal{F}(S_\pi)$. We argue that the optimal scanning policy must be *content-aware*. To formalize this principle, we decompose the problem into two coupled subproblems. **(1) Patch Importance Quantification:** Define an importance function $f_{\text{info}} : \mathcal{P} \to \mathbb{R}^+$ that assigns a scalar value $\mathcal{I}_k = f_{\text{info}}(p_k)$ to each patch based on its content, measuring its informativeness. **(2) Policy Optimization:** Given the importance scores $\{\mathcal{I}_k\}$, find the policy $\pi^*$ that maximizes the cumulative discounted information gain—a more informed objective:

$$\pi^* = \arg\max_{\pi \in \Pi} \sum_{t=1}^{N} \gamma^{t-1} \mathcal{I}_{s_t}, \tag{1}$$

where $s_t$ denotes the patch selected at step $t$ under policy $\pi$, and $\gamma \in (0, 1]$ is a discount factor that prioritizes early acquisition of high-information regions.

To achieve this information-maximizing scanning policy, we propose an information-gain-driven novel Vision model: InfoScan, which is built upon Visual Information State Space (VISS) blocks as illustrated in Figure 2. Each VISS block consists of a single network branch and two residual modules. In contrast to the standard VSS block, we replace the SS2D component with a Patch Selection Block (PSB), an Information Scoring Module (ISM), and a Path Planning Module (PPM) (see Figure 2); further details are provided in the next section. If not specified, all results reported in this paper are obtained using InfoScan models instantiated with this architecture.

### 3.1 INFORMATION SCORING MODULE

The Information Scoring Module provides a content-aware prior for adaptive scanning by quantifying the information content of image patches. We propose a composite score $\mathcal{I}(S)$ that jointly models global color diversity and local texture complexity:

$$\mathcal{I}(S) = \omega_1 \hat{H} + \omega_2 \hat{V}, \quad \text{where } \omega_1, \omega_2 \geq 0, \ \omega_1 + \omega_2 = 1. \tag{2}$$

Here, $\hat{H}$ and $\hat{V}$ denote the zero-mean unit-variance normalized Shannon entropy and local variance, respectively. The weights $\omega_1$ and $\omega_2$ are determined via grid search on the ImageNet-1K validation set to maximize classification accuracy under a fixed scanning budget, and are then fixed across all downstream experiments. We find the optimal setting to be $\omega_1 = 0.6$, $\omega_2 = 0.4$, indicating that both global and local cues contribute meaningfully, with a slight bias toward color diversity (The feasibility of $\omega_1 = 0.6$, $\omega_2 = 0.4$ on other vision tasks is validated in the Appendix C.1).

**Shannon Entropy (Global Diversity).** We compute the entropy of quantized color distributions to measure global color variety (Bromiley et al., 2004). Each RGB channel is uniformly quantized into $C$ bins, resulting in $K = C^3$ discrete levels. Let $p_k$ denote the empirical frequency of bin $k$. The entropy is $H = -\sum_{k=1}^{K} p_k \log p_k$ (with $0 \log 0 \triangleq 0$), which is then standardized across the dataset to obtain $\hat{H}$. Higher values indicate greater chromatic variation.

**Local Variance (Texture Complexity).** To capture fine-grained structure, we compute local intensity variance within $3 \times 3$ neighborhoods. For a pixel $(x, y)$ with neighborhood $N(x, y)$, the mean color is $\bar{I}_N = \frac{1}{|N|} \sum_{(u,v) \in N(x,y)} I(u, v)$, and the local variance is $V(x, y) = \frac{1}{|N|} \sum_{(u,v) \in N(x,y)} \|I(u, v) - \bar{I}_N\|^2$, where $\| \cdot \|^2$ denotes the squared Euclidean norm. The patch-level variance $V = \frac{1}{n^2} \sum_{(x,y) \in S} V(x, y)$ is standardized to yield $\hat{V}$.

**Boundary Information.** We further model inter-patch transitions by defining boundary salience. For a boundary $e$ between patches $S_1$ and $S_2$, we define $\mathcal{I}_b(e) = \mathcal{I}(S_1) \cdot \mathcal{I}(S_2)$, encouraging scanning paths to traverse between high-information regions and enhance contextual coherence in sequential processing.

## 3.2 Patch Selection Model

Let $N_p$ denote the side length of square image patches (i.e., patch size is $N_p \times N_p$). When dividing an image into such patches, there is a fundamental trade-off: smaller $N_p$ disrupts spatial context and incurs high computational overhead due to the generation of numerous patches; larger $N_p$, on the other hand, risks losing fine-grained details and increases processing latency per patch. To balance efficiency and information fidelity, we select the optimal patch size $N_p^\star$ by minimizing the total cost function $\mathcal{C}_{\text{total}}(N_p)$:

$$\mathcal{C}_{\text{total}}(N_p) = \lambda \mathcal{C}_e(N_p) + (1 - \lambda)\mathcal{C}_{\text{info}}(N_p), \quad \lambda \in [0, 1], \tag{3}$$

where $I = WH$ is the total number of pixels in the image, $N = I/N_p^2$ is the number of patches, $\mathcal{C}_e(N_p)$ measures efficiency-related costs, $\mathcal{C}_{\text{info}}(N_p)$ quantifies information loss, $T_{\text{total}}(N_p) = N \cdot T_{\text{patch}}(N_p)$ represents the total time required to scan the entire image, and $T_{\text{patch}}(N_p)$ is the time needed to process one $N_p \times N_p$ patch.

**Efficiency Term.** We model the delay per patch using a power-law model fitted on a calibration dataset $\mathcal{D}_{\text{calib}}$. Specifically, $\mathcal{D}_{\text{calib}}$ consists of 50K images from ImageNet-1K Val set, covering multiple classes to ensure diversity. For each image $x_i \in \mathcal{D}_{\text{calib}}$, we measure the execution times $y_i$ under different $N_p$ settings on the target hardware. This leads to the model: $T_{\text{patch}}(N_p) = k_p \cdot N_p^\alpha$ and $\mathcal{C}_e(N_p) = N \cdot T_{\text{patch}}(N_p) = k_1 I \cdot N_p^{\alpha-2}$, where $k_1 := k_p$. Here, $\alpha$ reflects the effective time complexity of patch processing. When $\alpha > 2$, the efficiency cost $\mathcal{C}_e(N_p)$ increases with $N_p$; it remains constant when $\alpha = 2$; and decreases when $\alpha < 2$.

**Information Term.** We model information loss as a U-shaped function, capturing the dual risk of insufficient global context at small $N_p$ and lost local details at large $N_p$:

$$\mathcal{C}_{\text{info}}(N_p) = \frac{k_2}{N_p^\beta} + k_3 N_p^\gamma, \qquad k_2, k_3 > 0, \ \beta, \gamma > 0. \tag{4}$$

The first term decays with increasing $N_p$ (indicating more complete global context), while the second term grows with $N_p$ (indicating worse local resolution).

**Optimization and Solution Strategy.** The optimal patch size is given by:

$$N_p^\star = \arg\min_{N_p} \left[ \lambda k_1 I N_p^{\alpha-2} + (1 - \lambda) \left( \frac{k_2}{N_p^\beta} + k_3 N_p^\gamma \right) \right]. \tag{5}$$

We solve this numerically: first, identify an interval containing the minimum over the feasible set $S$ (determined by image dimensions, stride constraints, and memory limits); then apply golden-section search in the continuous relaxation space; finally, round the result to the nearest valid $N_p \in S$. The trade-off parameter $\lambda$ is calibrated once to meet preset latency or memory budgets and remains fixed in subsequent experiments. Complete algorithmic details are provided in Appendix A.2. Notably, the determination of parameters $k_1$, $k_2$, and $k_3$ is detailed in Appendix A.4.

## 3.3 Path Planning Module

We reframe image scanning as an adaptive sequential decision process, moving beyond fixed, content-agnostic paths (e.g., raster, zigzag, or space-filling curves). Our core idea is to model the

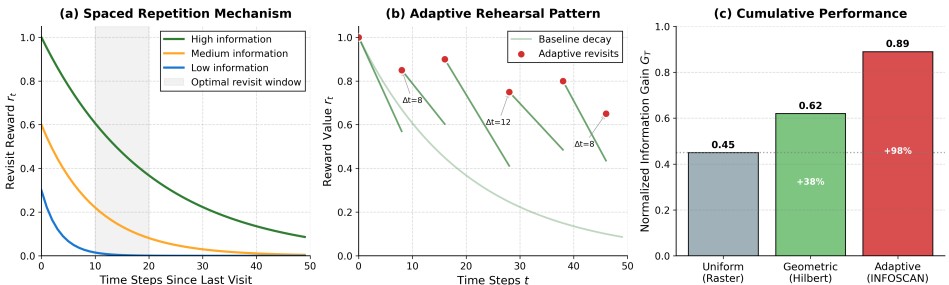

Figure 4: (a) Spatial repetition enables priority decay with reward attenuation for high-information regions, (b) dynamic rehearsal adjusts revisit intervals based on content importance, and (c) demonstrates that the proposed method achieves a significant 98% improvement in cumulative information gain compared to traditional scanning strategies.

scanner as an agent traversing the image plane, where the scanning path is dynamically shaped by the underlying content.

Formally, we partition the image into an $n \times n$ grid of patches, with each patch indexed by its coordinates $(i, j)$, where $i, j \in \{1, 2, \ldots, n\}$. The state space is defined as:

$$\mathcal{S} = \{(i, j) \mid i, j \in \{1, 2, \ldots, n\}\}. \tag{6}$$

The scanning process is modeled as a trajectory $\tau = (s_0, s_1, \ldots, s_T)$, where $s_t \in \mathcal{S}$ denotes the agent's location at time $t$, starting from an initial patch $s_0$.

Unlike traditional random walks with uniform transition probabilities, we formulate scanning as a *guided* random walk (Zhu & Ghahramani, 2002), which can be formalized as a Markov Decision Process (MDP) $\mathcal{M} = (\mathcal{S}, \mathcal{A}, P, r, \gamma)$:

(**State:**)$s_t = (i_t, j_t)$, optionally augmented with a visitation map $V_t \in \{0, 1\}^{n \times n}$, where $V_t(i, j) = 1$ if patch $(i, j)$ has been visited before $t$.

(**Action:**) $\mathcal{A} = \{\uparrow, \downarrow, \leftarrow, \rightarrow\}$ (4-connectivity), so $a_t \in \mathcal{A}(s_t)$ moves the agent to a neighboring patch.

(**Transition:**)Under a policy $\pi_\theta(a_t \mid s_t)$, the next state is deterministic: $s_{t+1} = f(s_t, a_t)$.

(**Discount:**) $\gamma \in [0, 1]$ weights immediate rewards more heavily.

The goal is to learn a policy, the details of the policy model are provided in Appendix A.5 $\pi_\theta$ that maximizes the expected cumulative return:

$$\max_\theta \mathbb{E}_{\tau \sim \pi_\theta} [G] = \mathbb{E}_{\pi_\theta} \left[ \sum_{t=0}^{T-1} \gamma^t r(s_t, a_t, s_{t+1}) \right]. \tag{7}$$

Markov Decision Process unifies two key objectives: (1) *exploration* of unvisited regions, and (2) *exploitation* of semantically rich areas.

### 3.4 REWARD-DRIVEN SCANNING

Adaptively discovering the optimal scanning path requires a reward mechanism that dynamically prioritizes information-rich regions while ensuring broad coverage. Our design is inspired by cognitive principles in human learning—specifically Levin (1986), *spaced repetition* and *focused rehearsal*—where important stimuli are revisited over time to strengthen perception and memory. To emulate this behavior, we introduce a content-adaptive reward function that slows the decay of revisit incentives for semantically salient patches, enabling periodic re-scanning while avoiding redundant fixations.

At each step, the model receives a reward that balances revisiting informative regions and exploring unvisited areas. Let $I(s) \in \mathbb{R}^+$ denote the information content of patch $s$. Let $k(s_{t+1})$ be the number of times patch $s_{t+1}$ has been visited within the past $t$ steps (with $k = 0$ if never visited). The decay factor $\alpha \in (0, 1)$ is *content-adaptive*: we set $\alpha = \alpha_{\text{high}}$ if $I(s) > \theta$, and $\alpha = \alpha_{\text{low}} < \alpha_{\text{high}}$ otherwise. This ensures that high-salience regions are "remembered" longer, promoting sustained

yet sparse revisits. The visitation indicator $V_t(s) \in \{0,1\}$ is 1 if patch $s$ has been observed before time $t$, and 0 otherwise. The weight $\lambda > 0$ controls the exploration bonus.

The reward function is then defined as:

$$r(s_t, a_t, s_{t+1}) = \underbrace{I(s_{t+1}) \cdot \alpha^{k(s_{t+1})}}_{\text{adaptive revisitation incentive}} + \underbrace{\lambda \cdot \left(1 - V_t(s_{t+1})\right)}_{\text{exploration bonus}} + \underbrace{\beta \cdot N_{\text{visited}}(s_{t+1})}_{\text{neighborhood information gain}} . \quad (8)$$

The adaptive revisitation incentive aligns with human-like visual behavior—repeatedly attending to meaningful content—while the exploration bonus ensures systematic scanning across the entire image. As shown in the Figure 4, Our reward design captures 98% of the achievable information gain, demonstrating high efficiency in perceptual resource allocation.

## 4 EXPERIMENTS

In this section, we conduct a series of experiments to evaluate the performance of InfoScan across various vision tasks and compare it with mainstream baseline models. The results are presented in the accompanying Figure 1. We further validate the effectiveness of each component in the proposed scanning strategy. All experiments are conducted on a server equipped with 16 V100 GPUs.

### 4.1 IMAGE CLASSIFICATION

**Settings:** We evaluate on ImageNet-1K (1.28M train, 50K val) (Deng et al., 2009) using $224 \times 224$ resolution. We primarily adopt the hyperparameter settings and experimental configurations from VMamba. During training, we use AdamW optimizer, cosine decay (initial LR $1 \times 10^{-3}$), and standard augmentations (e.g., random crop, flip, label smoothing).

**Results:** As shown in Table 1a, InfoScan outperforms CNN (Liu et al., 2022b; Koonce, 2021) Transformer, and SSM baselines in accuracy and efficiency. InfoScan-S achieves 84.64% top-1 accuracy, surpassing DeiT-S (Touvron et al., 2021a) (+4.0%) with fewer parameters. Compared to VMamba-S (83.24%), InfoScan-S gains +1.4% accuracy with 26M fewer parameters. These results demonstrate that information-aware dynamic scanning enables stronger and more efficient visual representation learning.

### 4.2 OBJECT DETECTION

**Setting:** We conduct experiments for object detection on the MSCOCO2017 (Lin et al., 2014) dataset, and we provide detailed experimental settings in the Appendix B.2.

**Results:** Table 1b demonstrates that InfoScan achieves competitive segmentation performance while maintaining a good parameter efficiency. It's worthy note that InfoScan-B achieves the highest $AP^{box}$ (49.8 %) and $AP^{mask}$ (44.7 %) scores among all methods tested, while using significantly fewer parameters (78M) compared to the other high-performing backbones like Swin-B (107M) and ConvNeXt-B (108M). The results verify InfoScan's effectiveness for detection tasks, which also implies that the information-aware scanning mechanism can be transfered well to detection tasks, not only classification and segmentation.

### 4.3 SEMANTIC SEGMENTATION

**Settings:** We evaluate semantic segmentation on ADE 20K (Zhou et al., 2017) and BraTS-2021 (Baid et al., 2021) using UperNet at $512 \times 512$. The training process follows standard protocols, and full details can be found in Appendix B.1. Notably, we additionally conducted generalization experiments on BraTS-2021 and ADE-20K. The experimental results and detailed analysis can be found in the Appendix C.2.

**Table 2:** Semantic segmentation performance on ADE-20K and BraTS-2021 with UPerNet at $512 \times 512$. P:Params; F:FLops.

| Backbone | ADE-20K | | BraTS-2021 | |
|---|---|---|---|---|
| | mIoU | P(M) | mIoU | P(M) |
| ResNet-50 | 41.9 | 67 | 17.4 | 67 |
| ResNet-101 | 45.2 | 86 | 20.7 | 86 |
| DeiT-T | 38.3 | 11 | 14.3 | 11 |
| DeiT-S | 42.4 | 43 | 21.3 | 43 |
| Vim-T | 40.3 | 13 | 18.7 | 13 |
| Vim-S | 45.3 | 46 | 20.4 | 46 |
| InfoScan-T | 40.9 | 10 | 19.3 | 10 |
| InfoScan-S | **45.8** | **38** | **22.3** | **38** |

Table 1: (a) ImageNet-1K classification at $224^2$ input (DeiT-B$^*$ at $384^2$). Throughput: per-GPU img/s. (b) Mask R-CNN on MSCOCO2017 val ($512 \times 2048$). AP$^b$/AP$^m$: box/mask AP; Sch.: schedule; MS: multi-scale; P: Params (M); F: FLOPs (G).

**(a) Image Classification**

| Method | P(M) | F(G) | Thr./Train | Acc(%) |
|---|---|---|---|---|
| DeiT-S | 22 | 4.6 | 96.4/137.3 | 74.70 |
| DeiT-B | 85 | 17.4 | 24.8/55.9 | 80.11 |
| DeiT-B$^*$ | 86 | 55.3 | 18.7/20.1 | 83.23 |
| Swin-T | 28 | 4.6 | 68.6/59.8 | 81.60 |
| Swin-S | 50 | 8.7 | 40.4/35.8 | 83.23 |
| Swin-B | 88 | 15.4 | 20.6/18.7 | 83.91 |
| VMamba-T | 31 | 4.9 | 77.2/24.8 | 82.47 |
| VMamba-S | 50 | 8.7 | 47.1/17.0 | 83.24 |
| VMamba-B | 89 | 15.4 | 29.4/12.2 | 84.32 |
| **InfoScan-T** | **10** | **2.5** | **91.7/46.5** | **83.43** |
| **InfoScan-S** | **24** | **4.8** | **63.8/33.6** | **84.64** |
| **InfoScan-B** | **38** | **8.4** | **54.0/26.2** | **85.19** |

**(b) Object Detection**

| Backbone | Sch. | AP$^b$ | AP$^m$ | P(M) |
|---|---|---|---|---|
| Swin-T | 1× | 42.9 | 38.1 | 48 |
| Swin-S | 1× | 44.5 | 39.6 | 69 |
| Swin-B | 1× | 45.6 | 41.5 | 107 |
| Swin-T | 3×MS | 46.5 | 41.5 | 48 |
| ConvNeXt-T | 1× | 44.3 | 39.6 | 48 |
| ConvNeXt-S | 1× | 45.5 | 40.9 | 70 |
| ConvNeXt-B | 1× | 47.2 | 42.4 | 108 |
| ConvNeXt-T | 3×MS | 46.1 | 41.3 | 48 |
| VMamba-T | 1× | 46.8 | 42.9 | 50 |
| VMamba-S | 1× | 48.4 | 43.3 | 70 |
| VMamba-B | 1× | 49.1 | 44.3 | 108 |
| VMamba-T | 3×MS | 48.3 | 43.6 | 50 |
| **InfoScan-T** | 1× | 47.1 | 41.2 | 42 |
| **InfoScan-S** | 1× | 48.8 | 43.4 | 59 |
| **InfoScan-B** | 1× | **49.8** | **44.7** | 78 |
| **InfoScan-T** | 3×MS | 48.6 | 43.2 | 42 |

**Results:** As shown in Table 2, INFOSCAN achieves superior accuracy-efficiency trade-offs. On ADE 20K, INFOSCAN-TI obtains 40.9% mIoU (vs. DeiT-Ti: 38.3%) with fewer parameters, and INFOSCAN-S reaches 45.8% (+0.5% over Vim-S) with 38M params. On BraTS-2021, it achieves 19.3% (TI) and 22.3% (S), outperforming DeiT and Vim. Notably, InfoScan-S matches UperNet–ResNet-101 (45.2%/20.7%) on both datasets with 56% fewer parameters. These results demonstrate strong generalization across natural and medical images.

## 4.4 ABLATION STUDY

We first conduct ablation experiments on the path scanning module and the patch planning module, using Vision Mamba as our baseline model for comparison. We fix the input image size to $512 \times 512$ and evaluate on three datasets. As shown in Table 3, on ImageNet-1K, the path scanning module improves InfoScan's accuracy from the baseline 82.5% to 83.4%, and the patch selection module further increases it to 85.9%.

Table 3: Ablation on core modules. E1 (Patch Selection), E2 (Path Planning).

| E1 | E2 | ImageNet-1K Top-1 Acc (%) | ADE-20K mIoU (%) | BraTS-2021 mIoU (%) |
|---|---|---|---|---|
| ✗ | ✗ | 82.5 | 45.3 | 18.7 |
| ✓ | ✗ | 83.4 | 45.7 | 18.9 |
| ✓ | ✓ | 85.9 | 45.9 | 19.3 |

We conduct ablation experiments on the three components of the reward function, using Vision Mamba as the baseline model. We fix the input image resolution to 512×512 and evaluate on three datasets. As shown in the Table 4, on ImageNet-1K, the combined effect of the adaptive revisitation incentive, exploration bonus, and neighborhood information gain boosts InfoScan's accuracy from 84.2% to 85.9%.

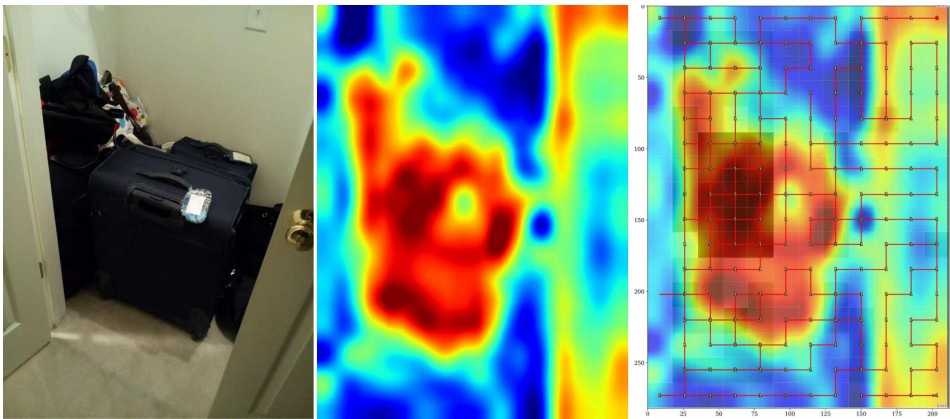

Figure 5: Visualization of the content-adaptive scanning path in InfoScan.

Notably, the ablation studies on Shannon Entropy and Local Variance in the Information Scoring Module are provided in the Appendix A.3.

Table 4: Ablation on reward components. M1 (revisitation), M2 (exploration), M3 (neighborhood gain).

| M1 | M2 | M3 | ImageNet-1K Top-1 Acc (%) | ADE-20K mIoU (%) | BraTS-2021 mIoU (%) |
|---|---|---|---|---|---|
| ✗ | ✗ | ✗ | 80.4 | 42.3 | 16.7 |
| ✓ | ✗ | ✗ | 81.1 | 42.7 | 17.8 |
| ✓ | ✓ | ✗ | 84.2 | 43.6 | 18.3 |
| ✓ | ✓ | ✓ | 85.9 | 45.9 | 19.1 |

## 4.5 SCANNING METHOD VISUAL COMPARISON

Figure 5 illustrates the visualization of the learned adaptive scanning path in InfoScan. From left to right: (1) the input image, (2) the information heatmap computed by combining Shannon entropy and local variance, where warmer colors (red/orange) indicate regions with higher complexity and richer content, and (3) the overlaid scanning trajectory on the heatmap. The red lines represent the adaptive scan path, which dynamically adjusts its density based on content importance: sparse scanning is applied in low-information areas (e.g., uniform walls), while dense, multi-pass traversal is adopted in complex regions (e.g., the suitcase and surrounding clutter). This demonstrates that InfoScan effectively prioritizes informative regions through a content-aware scanning strategy, achieving both efficiency and accuracy.

## 5 ANALYSES

A key question is whether performance gains stem from scan repetition or adaptive routing. To disentangle these factors, we conduct ablation studies with fixed scanning patterns under different configurations, as summarized in Table 5.

**Repetition yields diminishing returns.** Comparing single vs. triple passes of fixed patterns (ID 1 vs. ID 2, ID 3 vs. ID 5), we find that repeating scans brings marginal or even negative gains. Specifically, Triple Raster Scan (ID 2) achieves no improvement in Top-1 accuracy and degrades ADE-20K mIoU to 41.8%, likely due to redundant computation on low-salience regions. Triple Hilbert Scan (ID 5) improves ImageNet Top-1 by only +0.7% and ADE-20K mIoU by +0.6%, but reduces BraTS-2021 mIoU by −0.6%, indicating poor adaptation to structural heterogeneity in medical images.

**Stochastic initialization enhances coverage.** Randomizing the scan origin (ID 4) improves Single Hilbert Scan from $83.6\%$ to $84.5\%$ Top-1 and $42.9\%$ to $43.5\%$ mIoU on ADE-20K, confirming that random starts enhance spatial diversity. However, even with randomization, fixed-pattern methods plateau (e.g., ID 5), suggesting inherent limitations of static routing.

**Adaptive routing outperforms repetition and randomness.** InfoScan with random initialization (ID 7) achieves $85.9\%$ Top-1 and $45.9\%$ mIoU, surpassing all static baselines. Crucially, even with *fixed* initialization (ID 6), InfoScan ($84.2\%$, $44.6\%$) outperforms all ablated variants–including Triple Hilbert (ID 5) and the randomized Single Hilbert (ID 4)–demonstrating that learned adaptive routing is the primary driver of gains, not mechanical repetition or stochastic exploration.

**Learned policy enables intelligent revisitation.** These results confirm that performance improvements arise not from redundancy, but from *when and how* regions are revisited–guided by a reward-driven policy that balances exploration, uncertainty-based refinement, and local coherence. Intelligent policy design, rather than scan repetition, enables robust generalization across domains.

Table 5: Ablation study on scanning policy and starting point. "Fixed" denotes a predefined starting position; "Random" denotes a stochastically chosen start. InfoScan uses a reward-driven adaptive scanning policy with random initialization.

| ID | Method | Start | ImageNet-1K Top-1 (%) | ADE-20K mIoU (%) | BraTS-2021 mIoU (%) |
|----|--------|-------|------------------------|-------------------|----------------------|
| 1 | Single Raster Scan | Fixed | 83.5 | 42.9 | 15.3 |
| 2 | Triple Raster Scan | Fixed | 83.6 | 41.8 | 16.5 |
| 3 | Single Hilbert Scan | Fixed | 83.6 | 42.9 | 17.1 |
| 4 | Single Hilbert Scan | Random | 84.5 | 43.5 | 18.3 |
| 5 | Triple Hilbert Scan | Fixed | 84.8 | 44.2 | 16.8 |
| 6 | InfoScan | Fixed | 84.2 | 44.6 | 18.0 |
| 7 | InfoScan | Random | 85.9 | 45.9 | 19.3 |

# 6 CONCLUSION

This paper presents InfoScan, a novel visual backbone that introduces information-aware dynamic scanning for efficient high-resolution representation learning. By casting 2D spatial traversal as a sequential decision process guided by a learned salience metric, InfoScan departs from fixed or heuristic scanning patterns and adaptively allocates computation to informative regions. The core mechanism is integrated into a state space framework through the Visual Information State Space block, which supports flexible, content-dependent paths while maintaining near-linear complexity. Compared to prior state space and hierarchical vision models, InfoScan achieves improved efficiency and robust generalization across diverse vision tasks, including classification, dense prediction, and medical imaging. Notably, the learned scanning policy exhibits strong interpretability, aligning with human visual attention and enabling diagnostic analysis of model behavior. The design principle– scanning less but smarter–opens a new direction for efficient visual architectures beyond static token processing. In future work, we will adapt InfoScan to more vision tasks (Feng et al., 2025; Qian et al., 2025), and explore extending it to video modeling and adapting it to vision-language models (Liu et al., 2023; Bai et al., 2023).

ACKNOWLEDGEMENTS

This project was supported by National Natural Science Foundation of China (No. 62306132, 62402490, 62576225), Guangdong Basic and Applied Basic Research Foundation (No. 2025A1515011564), Natural Science Foundation of Shanghai (No. 25ZR1402136). We thank the anonymous reviewers for their insightful feedback on this work.

ETHICS STATEMENT

This work focuses on advancing the efficiency and adaptability of visual representation learning through algorithmic innovation. All experiments are conducted on publicly available datasets, in-

cluding ImageNet-1K, ADE-20K, MSCOCO2017, and BraTS-2021, which have been widely used in prior research under established ethical guidelines. We do not collect or use any private or sensitive data. The proposed method does not involve human subjects, personal information, or biometric identification. While the framework is general-purpose, potential misuse (e.g., in surveillance or deepfakes) is not specific to our approach and remains a broader concern for the machine learning community.

## REPRODUCIBILITY STATEMENT

We are committed to full reproducibility. All experimental details necessary to reproduce our results are provided in the main paper and the appendix, including model architectures, hyperparameters, training schedules, and optimization settings. We use standard benchmarks and publicly available datasets. The codebase, including training and evaluation scripts, will be released. Pre-trained models and detailed inference instructions will also be made publicly available. All experiments are conducted on standard hardware (16× NVIDIA V100 GPUs).

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

# A  MATHEMATICAL DERIVATIONS

## A.1  PATCH SIZE OPTIMIZATION DERIVATION

Here we provide the complete derivation of the optimal patch size formula from Section 3.1. Starting from the total cost function:

$$C_{\text{total}}(N_p) = \lambda k_1 I\, N_p^{\alpha-2} + (1-\lambda)\left(\frac{k_2}{N_p^{\beta}} + k_3 N_p^{\gamma}\right) \tag{9}$$

Taking the derivative with respect to $P$ and setting it to zero:

$$\frac{d}{dN_p}C_{\text{total}}(N_p) = \lambda k_1 I(\alpha-2)\, N_p^{\alpha-3} - (1-\lambda)k_2\beta\, N_p^{-\beta-1} + (1-\lambda)k_3\gamma\, N_p^{\gamma-1} = 0 \tag{10}$$

Multiplying through by $P^{\beta+1}$ to eliminate negative exponents:

$$\lambda k_1 I(\alpha-2)N_p^{\alpha+\beta-2} - (1-\lambda)k_2\beta + (1-\lambda)k_3\gamma N_p^{\gamma+\beta+1} = 0 \tag{11}$$

Let $m = \alpha + \beta - 2, n = \gamma + \beta + 1$ and

$$A = \lambda k_1 I(\alpha-2), B = (1-\lambda)k_2\beta, C = (1-\lambda)k_3\gamma,$$

we arrive at

$$AN_p^m + CN_p^n = B.$$

## A.2  INFORMATION THEORY FOUNDATIONS

**Normalization of Information Scores:** Given a dataset of patches $\{S_1, S_2, \ldots, S_N\}$, the normalization of Shannon entropy $H$ and local variance $V$ proceeds as follows:

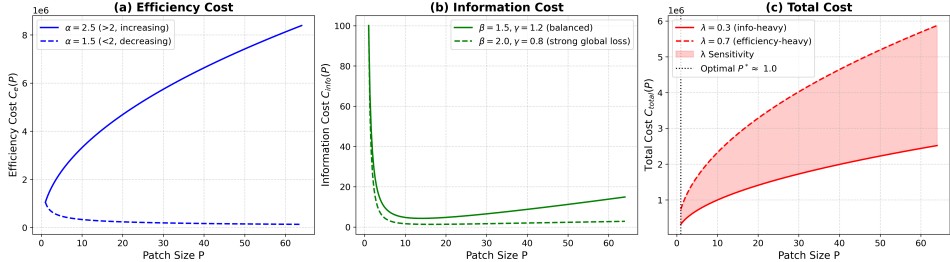

Figure 6: Patch size $P$ affects cost components differently: (a) efficiency cost increases with $P$, governed by exponent $\alpha$; (b) information cost drops sharply at small $p$, reflecting local redundancy; (c) total cost balances both, revealing an optimal $p^* \approx 1.0$ and sensitivity to trade-off weight $\lambda$.

For Shannon entropy:

$$\mu_H = \frac{1}{N} \sum_{i=1}^{N} H(S_i) \tag{12}$$

$$\sigma_H^2 = \frac{1}{N-1} \sum_{i=1}^{N} (H(S_i) - \mu_H)^2 \tag{13}$$

$$\hat{H}(S_i) = \frac{H(S_i) - \mu_H}{\sigma_H} \tag{14}$$

Similarly for local variance $V$. This ensures $\mathbb{E}[\hat{H}] = \mathbb{E}[\hat{V}] = 0$ and $\mathrm{Var}[\hat{H}] = \mathrm{Var}[\hat{V}] = 1$.

**Boundary Information Justification:** The boundary information $\mathcal{I}_b(e) = \mathcal{I}(S_1) \times \mathcal{I}(S_2)$ is motivated by the principle that transitions between high-information regions carry multiplicative importance. This captures the intuition that moving from one informative patch to another informative patch provides compound value for the scanning process.

### A.3 ANALYSES OF INFORMATION METRIC VALIDITY

To validate the necessity and effectiveness of our information metric, we conduct an ablation study on the components used to compute the information map: Shannon entropy (Q1) and local pixel variance (Q2). Results in Table 6 show that Shannon entropy component alone improves performance over the baseline without any information scoring (79.7% $\rightarrow$ 81.6% Top-1 on ImageNet-1K), confirming their individual utility in guiding adaptive scanning.

However, combining both Q1 and Q2 yields the best performance across all datasets—85.9% Top-1 accuracy on ImageNet-1K, +4.3% over Q1-only and +6.2% over no scoring—demonstrating that entropy and variance capture complementary aspects of visual information: global semantic diversity and local texture richness, respectively.

These results support the design of our composite information metric as not only empirically effective but also functionally justified. The significant gains in downstream tasks (e.g., +8.7% mIoU on BraTS-2021) further confirm that accurate information estimation is critical for efficient and adaptive vision modeling.

Table 6: Ablation on Information Scoring module. Q1 (Shannon Entropy), Q2 (Local Variance).

| Q1 | Q2 | ImageNet-1K Top-1 Acc (%) | ADE-20K mIoU (%) | BraTS-2021 mIoU (%) |
|----|----|---------------------------|------------------|---------------------|
| ✗ | ✗ | 79.7 | 37.2 | 12.1 |
| ✓ | ✗ | 81.6 | 43.2 | 17.2 |
| ✓ | ✓ | 85.9 | 45.9 | 19.3 |

A.4 PARAMETER ESTIMATION FOR THE PATCH SIZE OPTIMIZATION FRAMEWORK

We detail the procedure for estimating the model parameters $k_1$, $k_2$, and $k_3$ used in the total cost function:

$$\mathcal{C}_{\text{total}}(N_p) = \lambda \mathcal{C}_e(N_p) + (1 - \lambda)\mathcal{C}_{\text{info}}(N_p),$$

where $\mathcal{C}_e(N_p) = k_1 I \cdot N_p^{\alpha-2}$ and $\mathcal{C}_{\text{info}}(N_p) = \frac{k_2}{N_p^{\beta}} + k_3 N_p^{\gamma}$.

$k_1$: **Efficiency scaling coefficient.** The parameter $k_1$ (equivalent to $k_p$ in the patch timing model) captures the hardware- and model-specific constant in the power-law relationship $T_{\text{patch}}(N_p) = k_p \cdot N_p^{\alpha}$. It is estimated via linear regression on log-transformed timing measurements from the calibration dataset $\mathcal{D}_{\text{calib}}$. Specifically, we collect execution times $\{y_i\}$ for varying $N_p$ on 50K images from ImageNet-1K Val, and fit:

$$\log T_{\text{patch}}(N_p) = \alpha \log N_p + \log k_p.$$

Using ordinary least squares, we obtain estimates for $\alpha$ and $\log k_p$, from which $k_1 = k_p$ is derived. This ensures $\mathcal{C}_e(N_p)$ accurately reflects empirical computational latency.

$k_2, k_3$: **Information loss coefficients.** To estimate $k_2$ and $k_3$, Let $\phi(x; N_p)$ denote the deep features extracted from an image $x$ using patch size $N_p$. We measure information loss as the deviation from a reference representation $\phi(x; N_p^{\text{ref}})$, where $N_p^{\text{ref}} = 8$ is chosen as a high-resolution baseline:

$$\mathcal{L}_{\text{info}}(x, N_p) = \left\| \phi(x; N_p) - \phi(x; N_p^{\text{ref}}) \right\|_2^2.$$

We compute $\mathcal{L}_{\text{info}}(x, N_p)$ across $\mathcal{D}_{\text{calib}}$ for multiple $N_p$ values and average to obtain empirical information loss $\bar{\mathcal{L}}(N_p)$.

We then fit the parametric model $\mathcal{C}_{\text{info}}(N_p) = k_2 N_p^{-\beta} + k_3 N_p^{\gamma}$ to $\bar{\mathcal{L}}(N_p)$ using non-linear least squares, the resulting $k_2$ and $k_3$ ensure that the information cost term reflects the U-shaped trade-off between global context and local detail preservation.

A.5 POLICY NETWORK ARCHITECTURE

The policy network in InfoScan is responsible for learning an adaptive scanning policy $\pi(a_t|s_t; \theta)$ that selects the next patch to process based on the current state $s_t$. The network is trained via reinforcement learning to maximize the cumulative discounted information gain.

**State Representation** The input state $s_t$ at step $t$ is a 4-channel tensor composed of:

- Current position $(i_t, j_t)$: encoded as two scalar maps where each spatial location $(i, j)$ is assigned the normalized coordinates $\left(\frac{i}{n}, \frac{j}{n}\right)$.
- Information Map InfoMap$_t$: the spatial map of information scores $I(p_{i,j})$ computed by the Information Scoring Module (ISM), normalized to $[0, 1]$.
- Visitation Map $V_t$: a binary map indicating which patches have been visited (1 if visited, 0 otherwise).

These four channels are concatenated to form the input tensor of shape $n \times n \times 4$.

**Network Structure** The policy network is a lightweight convolutional neural network (CNN) with the following layers:

1. Input: $n \times n \times 4$ state tensor.
2. Convolutional Layer 1: $3 \times 3$ kernel, 64 filters, ReLU activation, stride 1, padding 1.
3. Convolutional Layer 2: $3 \times 3$ kernel, 64 filters, ReLU activation, stride 1, padding 1.
4. Global Average Pooling: reduces spatial dimensions to $1 \times 1 \times 64$.
5. Fully Connected Layer: 64 units, ReLU activation.

6. Output Layer: 4 units (corresponding to actions $\{\uparrow, \downarrow, \leftarrow, \rightarrow\}$), followed by softmax to produce action probabilities.

This design ensures near-constant computational cost regardless of image resolution, as the final layers operate on a fixed-size vector.

**Action Masking**   To prevent out-of-bound moves, we apply action masking during inference: if the agent is at the image boundary (e.g., $i_t = 1$), the "up" action is masked (set to zero probability) before softmax.

**Training Procedure**   We train the scanning policy using Proximal Policy Optimization (PPO). The agent interacts with the image grid environment over episodes of fixed length $T = 10$ steps, corresponding to a sparse scan path across the image.

At each step $t$, the policy network takes as input the state tensor $s_t \in \mathbb{R}^{n \times n \times 4}$ and outputs a probability distribution over the four movement actions. An action $a_t$ is sampled from this distribution during training for exploration. After executing $a_t$, the agent transitions to patch $s_{t+1}$ and receives a reward $r_t = r(s_t, a_t, s_{t+1})$, composed of adaptive revisitation incentive, exploration bonus, and neighborhood gain.

The total objective maximizes the discounted cumulative reward:

$$ J(\theta) = \mathbb{E}_{\tau \sim \pi_\theta} \left[ \sum_{t=0}^{T} \gamma^t r_t \right], $$

with discount factor $\gamma = 0.99$. To stabilize training:

- **Reward normalization**: We maintain a moving average of recent rewards and normalize each $r_t$ online using zero-mean, unit-variance scaling.
- **Entropy regularization**: We include an entropy bonus $\mathcal{L}_{\text{entropy}}$ to encourage exploration in early stages.
- **Gradient clipping**: Global norm clipped at 0.5.

We use the Adam optimizer with learning rate $3 \times 10^{-4}$, batch size 64 (aggregated over 8 parallel environments), and update the policy every $T = 10$ steps using 3 PPO epochs per update. The value head (an additional output branch from the penultimate FC layer) is trained jointly to estimate state value $V(s_t)$, with coefficient $\lambda_v = 0.5$ balancing the value loss. Training runs for 200K iterations on ImageNet-1K training set images resized to $512 \times 512$.

## B   EXPERIMENTAL SETUP DETAILS

### B.1   SETTINGS FOR SEMANTIC SEGMENTATION.

We conduct semantic segmentation experiments on the ADE-20K and BraTS-2021 datasets. ADE-20K contains 150 fine-grained semantic categories, with 20K images for training, 2K for validation, and 3K for testing. BraTS-2021 includes three semantic classes (tumor sub-regions), and we use T1-weighted MRI scans as input. The training, validation, and test sets contain 21K, 3K, and 6K slices, respectively. We adopt UperNet as the base framework. During training, we use the AdamW optimizer with a weight decay of 0.01 and a total batch size of 24. The learning rate is initialized to $8 \times 10^{-5}$, decayed linearly, and warmed up over the first 2,000 iterations. Total training runs for 180K iterations. Standard data augmentations are applied: random horizontal flipping, random rescaling within the range $[0.5, 2.0]$, and random photometric distortion. At evaluation, input images are resized such that the shorter side is 512 pixels.

### B.2   SETTINGS FOR OBJECT DETECTION.

We conduct object detection experiments on the MS-COCO 2017 dataset. The dataset contains 118K training images, 5K validation images, and 20K test images. We adopt the standard Cascade

Mask R-CNN as the base framework. For ViT-based backbones, we follow ViTDet and apply additional designs—such as interleaved window and global attention—to handle high-resolution inputs. For SSM-based Vim backbones, we use the original architecture without modifications. All other training and evaluation settings remain consistent across variants. We optimize using AdamW with a weight decay of 0.1 and a total batch size of 64. The learning rate is initialized to $1 \times 10^{-4}$ and decayed linearly over 380K iterations.

## C  GENERALIZATION EXPERIMENTS

### C.1  FEASIBILITY OF THE INFORMATION SCORING WEIGHT IN OTHER TASKS

The weights $\omega_1$ and $\omega_2$ in the information scoring module are determined via grid search on the ImageNet-1K validation set to maximize classification accuracy under a fixed computational budget. To validate the generalization of this optimal setting to other vision tasks, we evaluate the performance of InfoScan on two distinct downstream tasks—semantic segmentation on ADE-20K and medical image segmentation on BraTS-2021—using the same fixed weights, without task-specific re-tuning.

Results in Table 7 demonstrate that the weight configuration optimized on ImageNet-1K ($\omega_1 = 0.6, \omega_2 = 0.4$) consistently yields the best performance across both datasets. This indicates strong transferability of the information scoring mechanism, suggesting that the relative importance of global color diversity ($\omega_1$) and local texture complexity ($\omega_2$) learned from natural images generalizes well to both complex scene parsing and fine-grained medical analysis.

Table 7: Ablation on information scoring weights $\omega_1$ (entropy) and $\omega_2$ (variance) evaluated on ADE-20K and BraTS-2021. The optimal weights ($\omega_1 = 0.6, \omega_2 = 0.4$) selected on ImageNet-1K achieve the highest mIoU on both tasks, confirming their cross-task effectiveness.

| $\omega_1$ | $\omega_2$ | ADE-20K mIoU (%) | BraTS-2021 mIoU (%) |
|---|---|---|---|
| 0.5 | 0.5 | 36.7 | 18.7 |
| 0.4 | 0.6 | 42.7 | 20.4 |
| 0.6 | 0.4 | **45.8** | **22.3** |

### C.2  GENERALIZATION EXPERIMENTS ON SEGMENTATION TASKS

We conduct cross-task and cross-domain generalization experiments in semantic segmentation. Specifically, we test whether models trained on one segmentation dataset can generalize to a significantly different one *without fine-tuning*, simulating real-world deployment where target-domain labels are unavailable.

We compare InfoScan-T with standard CNN (ResNet-50) and recent vision architectures (DeiT-T, Vim-T) under a zero-shot domain transfer setting. All models use the same segmentation head UPerNet, are trained on one dataset, and directly evaluated on the other. Input resolution is fixed at $512 \times 512$. The two datasets represent highly distinct domains: Results are reported in Table 8, measured by mean Intersection-over-Union (mIoU).

## D  LIMITATIONS

(i) First, the current implementation assumes a uniform patch size and grid structure, which may not optimally capture multi-scale semantics in high-resolution images. Future work could explore adaptive patching or hierarchical scanning strategies.

(ii)Second, the information scoring module, though lightweight, introduces additional latency during inference. end-to-end deployment in real-time systems requires further optimization of the scoring and policy inference pipeline.

(iii)Third, the reward function contains hyperparameters (e.g., $\alpha_{\text{high}}, \lambda$) that currently require mild tuning for extreme domain shifts (e.g., natural to medical). Although we observe consistent rank-

Table 8: Zero-shot cross-dataset generalization performance on semantic segmentation tasks. Models are trained on one dataset and evaluated on the other without fine-tuning. InfoScan-T shows superior transferability, achieving higher mIoU in both directions, indicating stronger generalization to unseen domains and modalities.

| Backbone | Train Dataset | Test Dataset | mIoU (%) |
|---|---|---|---|
| ResNet-50 | ADE-20K | BraTS-2021 | 4.9 |
| ResNet-50 | BraTS-2021 | ADE-20K | 5.3 |
| DeiT-T | ADE-20K | BraTS-2021 | 8.7 |
| DeiT-T | BraTS-2021 | ADE-20K | 8.2 |
| Vim-T | ADE-20K | BraTS-2021 | 7.8 |
| Vim-T | BraTS-2021 | ADE-20K | 7.9 ' |
| InfoScan-T | ADE-20K | BraTS-2021 | **11.4** |
| InfoScan-T | BraTS-2021 | ADE-20K | **11.2** |

ing across settings, fully automatic adaptation without any validation feedback remains an open challenge.

(iv)Finally, all experiments focus on 2D images; extension to 3D Videos would require re-designing the action space and scanning policy, which we leave for future work.

## E    LLM USAGE

The manuscript was polished using a large language model (LLM). After revision, the methodological and experimental details were verified and confirmed by the authors.

