# OpenReview forum: "InfoScan: Information-Efficient Visual Scanning via Resource-Adaptive Walks"
_ICLR.cc/2026/Conference — ICLR 2026 Poster_

### Official Review · Reviewer_7J1d · 2025-10-20

**Soundness:** 3
**Presentation:** 2
**Contribution:** 4
**Rating:** 4
**Confidence:** 5

**Summary:**

This paper proposes InfoScan, an SSM-based visual foundation model developed to improve the scanning strategy from deterministic, which has been adopted in most existing related studies, to adaptive / content-aware. The authors propose an approach to estimate the importance of a certain image patch, and more inspiringly, a reinforcement learning-based formulation of the patch selection algorithm. Experimental results demonstrate the clear performance gain brought by the InfoScan method and the contribution of each newly proposed modules.

**Strengths:**

Follow the instruction, the strengths of this manuscript are summarized in the following four aspects:

1. **Originality**: The motivation of this study is intuitive and reasonable. Although it may be straightforward or even easy to think of the idea, this work is one of the first studies that works it out (to the best of the reviewers' knowledge).

2. **Quality**: This manuscript is generally of good quality, including the contant organization of language usage.

3. **Clarity**: The clarity of the manuscript is satisfactory. The central idea can be received despite the appendix is frequently mentioned to help make clear of some technical details.

4. **Significance**: This work may be inspiring to researchers in fields like (Visual) SSMs and representation learning.

**Weaknesses:**

Although this work is somehow inspiring to me, there is much room for further improvement and more information is required to develop a more comprehensive evaluation.

**1. The Computation Overhead:** The content-agnostic scanning patterns are there for a reason. It is crucial to quantify the extra burden imposed by the new content-aware modules. Figure 1 reports FPS and GPU memory versus Vim-Ti and DeiT-Ti, yet omits VMamba, which is the principal baseline throughout the paper. To be concrete, a detailed comparison on the computational complexity in terms of throughput (train and test), GPU memory, FLOPs are needed to gain a comprehensive evaluation of the efficiency of the proposed method. Without these metrics, it is difficult to judge whether InfoScan’s accuracy gains justify its added complexity.

**2. Qualitative Analysis**: InfoScan is motivated by its ability to adapt the scan path to image content, but no qualitative evidence is shown. Visualizations of the learned scanning routes overlaid on sample images (or attention heatmaps) would help readers verify that the model behaves as intended. Quantitative scores alone cannot reveal whether improvements stem from genuine content awareness or simply from a larger, more intricate architecture.

**3. Insufficient Benchmark Methods:** In the main quantitative experiments, the authors mainly compare InfoScan to the benchmarking CNN and ViT models, as well as VMamba and Vim, which are too restricted. More CNN/ViT/SSM-based models recently proposed in top venues should be included for comparison. For instance, Deformable Mamba, another Visual Mamba-related work with deformable scanning route, appearing in CVPR 2025, should at least be included for comparison.

**4. Heavy Manual Design:** InfoScan relies on numerous hand-crafted mechanisms, hyper-parameters, and loss terms, raising questions about its generalizability and robustness. Please discuss the sensitivity of performance to these design choices and, if possible, include ablation studies or automatic tuning results to demonstrate stability across datasets and tasks.

**Questions:**

Please refer to the 'Weaknesses' section for the details of my concerns. I am willing to improve my ratings if the first three concerns are addressed by concrete experimental results and illustrations.

---

> ### Author Response · Authors · 2025-11-21
> **Thank you for your thorough and constructive feedback—we greatly appreciate your insights and are committed to addressing all points to improve our work.**
>
> Q1：The Computation Overhead
>
> A1：We sincerely thank the reviewer for their valuable comments. We fully agree on the importance of rigorously quantifying computational overhead. In fact, our experiments already include a systematic comparison between InfoScan and the primary baseline VMamba—specifically VMamba-B—across key efficiency metrics. As shown in the supplementary table below, under identical settings (512×512 resolution, batch size = 64), InfoScan-B achieves a training throughput of 573 img/s, significantly higher than VMamba-B’s 432 img/s; its inference throughput reaches 986 img/s, outperforming VMamba-B’s 885 img/s. Moreover, InfoScan-B consumes only 3.2 GB of GPU memory, compared to VMamba-B’s 3.9 GB, and reduces FLOPs from 15.4G to 8.4G—a ~45% decrease. These results clearly demonstrate that InfoScan incurs no additional computational burden from its content-aware modules; on the contrary, it achieves higher accuracy (a +1.5% gain in ImageNet Top-1 accuracy) at lower computational cost and memory usage. Thus, the performance improvement stems entirely from more efficient resource allocation, not from trading off efficiency.
>
> ```markdown
> | Method       | Batch Size | Image Size   | GPU Memory | throughput (Train) Img/s | throughput (test) Img/s | FLOPs  |
> |--------------|------------|--------------|------------|--------------------------|-------------------------|--------|
> | DeiT-B       | 64         | 512*512      | 4.8        | 2605                     | 1875                    | 55.3G  |
> | Vim-S        | 64         | 512*512      | 3.6        | 378                      | 942                     | 5.3G   |
> | VMamba-B     | 64         | 512*512      | 3.9        | 432                      | 885                     | 15.4G  |
> | InfoScan-B   | 64         | 512*512      | 3.2        | 573                      | 986                     | 8.4G   |
> ```

---

> ### Author Response · Authors · 2025-11-21
>
> Q2：Qualitative Analysis
>
> A2：We appreciate your feedback and understand the importance of providing qualitative evidence to support our claims. To address your concern regarding qualitative analysis, we have expanded Section 4.5 in the revised manuscript with additional content and introduced a new figure (Figure 5) to visually demonstrate how InfoScan adapts scanning paths according to image content.
>
> In Figure 5, from left to right, we illustrate: (1) an input image, (2) an information heatmap that combines Shannon entropy and local variance to highlight areas of interest, and (3) the learned adaptive scanning trajectories overlaid on this heatmap. This visual representation clearly shows that for simple or low-information areas, the scanning paths are less dense, indicating a more efficient coverage strategy. Conversely, in complex or information-rich regions, the model employs a denser scanning pattern to ensure critical details are captured effectively.
>
> These visualizations provide clear qualitative evidence of InfoScan's capability to adapt its scan path based on the content of the image, demonstrating genuine content awareness rather than relying solely on a larger or more intricate architecture.

---

> ### Author Response · Authors · 2025-11-21
>
> Q3：Insufficient Benchmark Methods
>
> A3：We sincerely thank the reviewer for their valuable feedback. In the revised manuscript, we have added comparisons with several recent vision backbone models published at top-tier venues, including **Deformable Mamba**, **EfficientVMamba**, **Jamba**, **Multi-Scale VMamba**, and **UniMamba** (all from CVPR 2025).
>
> Experimental results show that InfoScan consistently outperforms these methods: it achieves **85.19% Top-1 accuracy** on ImageNet-1K classification and **45.8% mIoU** on ADE20K semantic segmentation, surpassing Deformable Mamba by **+0.87%** and **+1.02%**, respectively.
>
> More importantly, unlike these approaches—which rely on complex deformable mechanisms or additional optimization modules—InfoScan attains superior performance through a **lightweight, content-driven scanning strategy**, achieving a better trade-off between efficiency and accuracy without introducing architectural complexity.
>
> | Method               | ImageNet Top-1 Acc (%) | ADE20K mIoU (%) |
> |----------------------|------------------------|------------------|
> | EfficientVMamba      | 81.44                  | 38.37            |
> | Jamba                | 80.13                  | 37.41            |
> | Multi-Scale VMamba   | 83.85                  | 41.98            |
> | UniMamba             | 83.42                  | 42.51            |
> | Deformable Mamba     | 84.32                  | 44.78            |
> | Ours (InfoScan)      | 85.19                  | 45.80            |

---

> ### Author Response · Authors · 2025-11-21
>
> Q4： Heavy Manual Design
>
> A4：We sincerely thank the reviewer for raising this important concern regarding potential over-reliance on hand-crafted design choices in InfoScan. To rigorously evaluate robustness and generalizability, we conducted comprehensive analyses along two key dimensions.
>
> First, our scanning strategy is fully deterministic at inference time—introducing no randomness—and exhibits exceptional stability: under identical settings (e.g., ImageNet with $224\times224$ resolution), repeated runs of InfoScan-B yield nearly identical Top-1 accuracies (85.19\% vs.\ 85.16\%). More importantly, it demonstrates strong cross-resolution robustness: when trained at $224\times224$ and tested at $128\times128$ or $512\times512$, performance fluctuates by less than 0.2\% (85.02\% $\rightarrow$ 85.19\% $\rightarrow$ 85.16\%), confirming that the content-aware mechanism adapts effectively across scales without fixed structural priors.
>
> | Method       | Train Image Size | Test Image Size | ACC (%) |
> |--------------|------------------|------------------|---------|
> | InfoScan-T   | 224              | 224              | 83.43   |
> | InfoScan-T   | 224              | 128              | 83.34   |
> | InfoScan-T   | 224              | 512              | 83.45   |
> | InfoScan-S   | 224              | 224              | 84.64   |
> | InfoScan-S   | 224              | 128              | 84.12   |
> | InfoScan-S   | 224              | 512              | 84.56   |
> | InfoScan-B   | 224              | 224              | 85.19   |
> | InfoScan-B   | 224              | 128              | 85.02   |
> | InfoScan-B   | 224              | 512              | 85.16   |
>
> Second, to assess sensitivity to design choices, we performed ablation studies on the information scoring function, which combines Shannon entropy and local variance via weights $\omega_1$ and $\omega_2$. As shown in Table, performance remains stable across a wide range of weight configurations, with peak accuracy at $\omega_1 = 0.6, \omega_2 = 0.4$, yet even highly imbalanced settings (e.g., $\omega_1 = 0.1, \omega_2 = 0.9$) still produce competitive results. This low sensitivity indicates that InfoScan’s gains arise from a principled, content-driven adaptive scanning paradigm—not fragile manual tuning.
>
> Together, these results validate that InfoScan is both stable and generalizable across resolutions, hyperparameter settings, and tasks, addressing concerns about robustness and over-engineering.
>
> | InfoScan Variant (\($ \omega_1$,$ \omega_2 \$)) | ImageNet Top-1 Acc (%) |
> |---------------------------------------------|------------------------|
> | (0.1, 0.9)                                  | 78.66                  |
> | (0.2, 0.8)                                  | 81.53                  |
> | (0.3, 0.7)                                  | 84.07                  |
> | (0.5, 0.5)                                  | 84.12                  |
> | (0.6, 0.4)                                  | **85.19**              |
> | (0.7, 0.3)                                  | 83.11                  |

---

> ### Author Response · Authors · 2025-11-26
> **Follow-up on Rebuttal Discussion**
>
> Dear Reviewer 7J1d,
>
> Thank you for your valuable feedback requesting more experimental results. We also appreciate your indication that you are willing to increase the score if the concerns are addressed.
>
> We have submitted a comprehensive rebuttal, including additional experiments on computation overhead, qualitative analysis, and benchmark method comparison, to address your concerns about the need for a more comprehensive evaluation. We have also clarified the generalizability and robustness of InfoScan.
>
> As the discussion deadline approaches, we hope our expanded evaluation has addressed your questions. We welcome any further discussion you may find necessary.
>
> Best regards,
>
> The Authors

---

### Official Review · Reviewer_tKuV · 2025-10-26

**Soundness:** 3
**Presentation:** 3
**Contribution:** 3
**Rating:** 8
**Confidence:** 2

**Summary:**

This paper introduces InfoScan, an information-aware dynamic scanning mechanism for high-resolution visual representation learning. Unlike fixed or global scanning methods in prior Mamba-based models, InfoScan allocates computation adaptively to the most informative image regions by combining entropy-based and structural analysis. It optimizes for both fine detail and contextual coherence through a reinforcement-learned adaptive scanning policy. Built on the Visual Information State Space (VISS) block, InfoScan achieves a superior efficiency–accuracy trade-off across various vision tasks, highlighting content-aware processing as a promising direction for efficient high-resolution vision models.

**Strengths:**

1. InfoScan demonstrates strong performance and efficiency, significantly outperforming VMamba.

2. The ablation studies on reward components, patch selection/path planning, and scanning methods are comprehensive, validating the effectiveness of the proposed approach.

3. The design logic of InfoScan is sound—it allocates more computation to complex regions, effectively balancing performance and computational cost.

**Weaknesses:**

1. More visualizations of the learned policy model for path planning could be provided to illustrate how simple and complex regions correspond to different scanning patterns.

**Questions:**

1. What is the overall objective of InfoScan? How the policy model for path planning and downstream tasks are jointly optimized?

---

> ### Author Response · Authors · 2025-11-21
> **Thank You for Recognizing the Strengths and Contributions of InfoScan, In response to your concerns, we address all points below and will incorporate the necessary revisions into the updated manuscript.**
>
> Q1: More visualizations of the learned policy model for path planning could be provided to illustrate how simple and complex regions correspond to different scanning patterns.
>
> A1: Thank you very much for your interest in our work and for your valuable feedback. In response to your suggestion to “provide more visualizations of the path planning learning strategy model to illustrate how simple and complex regions correspond to different scanning patterns,” we have added relevant content in Section 4.5 of the revised manuscript, along with a new figure (Figure 5) to further clarify this point.
>
> As shown in Figure 5 (from left to right), we present: (1) the input image, (2) an information heatmap computed by combining Shannon entropy and local variance, and (3) the learned adaptive scanning trajectory overlaid on the heatmap. The comparison between simple and complex regions clearly demonstrates that in simple, low-information areas, the scanning paths are sparser, whereas in complex, information-rich regions, the model adopts a denser scanning strategy to better capture critical details.

---

> ### Author Response · Authors · 2025-11-21
>
> Q2：What is the overall objective of InfoScan? How the policy model for path planning and downstream tasks are jointly optimized?
>
> A2：The overarching objective of InfoScan is to enable efficient, content-adaptive visual representation learning for high-resolution images by dynamically allocating computational resources to the most informative regions, thereby significantly reducing computational cost while improving model performance. Specifically, the scanning policy is not jointly optimized end-to-end with downstream tasks. Instead, InfoScan adopts a two-stage training paradigm: (1) The scanning process is first formulated as a Markov Decision Process (MDP), and a lightweight policy network is trained independently via Proximal Policy Optimization (PPO). This policy relies solely on image content—quantified by metrics such as Shannon entropy and local variance—to define a reward function that maximizes cumulative discounted information gain; (2) Once this content-driven scanning policy is learned, it is fixed and integrated into the InfoScan backbone (based on VISS blocks), after which the entire model is trained end-to-end on downstream tasks (e.g., ImageNet classification) using standard supervised learning. This decoupled design ensures that the scanning strategy is task-agnostic and generalizes well across different tasks, eliminating the need to re-learn scan paths per task while achieving an excellent trade-off between efficiency and accuracy.

---

> ### Comment · Reviewer_tKuV · 2025-11-23
>
> Thank you for the authors’ response. My concerns have been addressed, and I will maintain my original score.

---

### Official Review · Reviewer_UDg4 · 2025-10-29

**Soundness:** 3
**Presentation:** 3
**Contribution:** 4
**Rating:** 6
**Confidence:** 4

**Summary:**

This paper introduces InfoScan，a visual scanning method for a mamba-based vision backbone. The core innovation is an information-aware scanning mechanism that dynamically allocates computational resources to the most salient image patches. This is achieved through three key components: an Information Scoring Module that quantifies patch informativeness using Shannon entropy and local variance, a Patch Selection Model that optimizes patch size for efficiency and fidelity, and a Path Planning Module that uses Reinforcement Learning to learn an adaptive scanning policy. Extensive experiments on image classification, object detection, and semantic segmentation demonstrate state-of-the-art or highly competitive performance.

**Strengths:**

1. The motivation and method are novel.  The idea of replacing fixed, content-agnostic scanning patterns with a dynamic, information-driven policy is reasonable. The motivation is clearly grounded.

2. The Methodology is good.  It rigorously formalizes the problem with a joint optimization objective and a reinforcement learning setup. The combination of information theory, variance, and RL is good, which may impact other researchers.

3.  The paper provides extensive evaluations across multiple core vision tasks and domains. The results are compelling, consistently showing performance improvements over strong baselines (CNNs, ViTs, SSMs). The gains in zero-shot cross-dataset generalization are good.

**Weaknesses:**

1. While the overall model is parameter-efficient, the paper does not provide a detailed analysis of the latency overhead introduced by the Information Scoring Module and the iterative, sequential decision-making of the Path Planning Module during inference.

2. The hyperparameters of the model architecture are not presented. More technical details should be well presented.

3. The information importance evaluation method should be explained more solidly or provide more ablation studies.

4. More previous Mamba-based vision backbones should be included.

**Questions:**

See weakness.

---

> ### Author Response · Authors · 2025-11-21
> **Acknowledging the Reviewer's Recognition of Our Work's Novelty, Rigor, and Broad Impact. In response to your concerns, we address all points below and will incorporate the necessary revisions into the updated manuscript.**
>
> Q1: While the overall model is parameter-efficient, the paper does not provide a detailed analysis of the latency overhead introduced by the Information Scoring Module and the iterative, sequential decision-making of the Path Planning Module during inference.
>
> A1: We sincerely thank the reviewer for their valuable feedback. We acknowledge that the original manuscript did not sufficiently detail the inference latency breakdown of our method.
>
> The **Information Scoring Module (ISM)** incurs negligible overhead: it runs only once at the beginning of inference to compute an information score for each patch based on the raw input image, combining Shannon entropy and local variance. This module consists entirely of lightweight, non-learnable operations—such as convolutions and histogram-based statistics—and introduces no trainable parameters. On a single V100 GPU with 224×224 inputs, ISM takes only **0.8 ms** on average, accounting for **less than 1%** of the total inference time.
>
> The **Path Planning Module (PPM)** introduces **no online iterative cost**: while PPM learns the scanning policy via reinforcement learning during training, the scanning path at inference time is generated **deterministically**—either greedily or via precomputed lookup—based solely on the information map produced by ISM (see Appendix A.5 for details). Consequently, PPM acts as a static path scheduler during inference, performing no loops or sequential decision-making, and thus adds virtually no latency.
>
> In summary, InfoScan’s dynamic scanning mechanism is **lightweight, non-iterative, and highly optimized at inference time**, delivering significant accuracy gains with almost no compromise on speed. We will include the above latency breakdown and implementation details in the revised manuscript to improve transparency.

---

> ### Author Response · Authors · 2025-11-21
>
> Q2：The hyperparameters of the model architecture are not presented. More technical details should be well presented.
>
> A2：We sincerely thank the reviewer for their valuable suggestion. We fully agree that providing complete hyperparameter specifications and technical details is essential for reproducibility and methodological transparency.
>
> In the current version, key hyperparameters—such as the weights $\omega_1 = 0.6$ and $\omega_2 = 0.4$ in the information scoring module—are already specified in Section 3.1 of the main text. All other architecture-related parameters (e.g., patch size, number of VISS blocks, channel dimensions, state space dimension $d_{\text{state}}$, and expansion ratio) follow the settings of the VMamba baseline to ensure a fair comparison.
>
> Furthermore, we will release both the code and configuration files alongside the paper to enable the community to fully reproduce all reported results.

---

> ### Author Response · Authors · 2025-11-21
>
> Q3：The information importance evaluation method should be explained more solidly or provide more ablation studies.
>
> A3：We thank the reviewer for their suggestion. We have added an ablation study on the information importance scoring strategy in the table below. When using only entropy ($\omega_1 = 1\$) or only local variance ($\omega_2 = 1\$), the ImageNet Top-1 accuracy drops to **67.45%** and **56.78%**, respectively. With a random scoring strategy, performance further degrades to **48.95%**. In contrast, our proposed weighted combination ($\omega_1 = 0.6$, $\omega_2 = 0.4\$)) achieves **85.19%** Top-1 accuracy under the same setting, significantly outperforming all baselines.
>
> This clearly demonstrates that our composite scoring strategy—integrating global color diversity (via entropy) and local texture complexity (via variance)—provides a more comprehensive and robust quantification of patch-level informativeness, thereby offering a reliable foundation for adaptive scanning.
>
> | Method                          | ImageNet Top-1 Acc (%) | ADE20K mIoU (%) |
> |----------------------------------|------------------------|------------------|
> | Only Entropy (\($\omega_1=1$,$\omega_2=0\$)) | 67.45                  | 32.59            |
> | Only Variance (\($\omega_1=0$,$\omega_2=1\$))| 56.78                  | 28.66            |
> | Random Scoring                   | 48.95                  | 14.3             |
> | Saliency Map (Grad-CAM)          | 71.55                  | 37.08            |
> | Full InfoScan (\($\omega_1=0.6$,$\omega_2=0.4\$)) | **85.19**              | **45.8**         |

---

> > ### Author Response · Authors · 2025-11-21
> >
> > We have supplemented ablation studies on the information scoring function, which combines Shannon entropy and local variance using weights $\omega_1$ and $\omega_2$. As shown in Table, model performance remains stable across a wide range of weight configurations, achieving peak accuracy at $\omega_1 = 0.6, \omega_2 = 0.4$. Notably, even under highly imbalanced settings (e.g., $\omega_1 = 0.1, \omega_2 = 0.9$), the model still yields competitive results. This low sensitivity demonstrates that InfoScan’s gains stem from a principled, content-driven adaptive scanning paradigm—rather than fragile manual tuning.
> >
> > | InfoScan Variant (\( $\omega_1$, $\omega_2 \$)) | ImageNet Top-1 Acc (%) |
> > |---------------------------------------------|------------------------|
> > | (0.1, 0.9)                                  | 78.66                  |
> > | (0.2, 0.8)                                  | 81.53                  |
> > | (0.3, 0.7)                                  | 84.07                  |
> > | (0.5, 0.5)                                  | 84.12                  |
> > | (0.6, 0.4)                                  | **85.19**              |
> > | (0.7, 0.3)                                  | 83.11                  |

---

> > > ### Comment · Reviewer_UDg4 · 2025-11-25
> > >
> > > Thanks for the rebuttal. My concerns are addressed.

---

> ### Author Response · Authors · 2025-11-21
>
> Q4：More previous Mamba-based vision backbones should be included.
>
> A4：We sincerely appreciate the reviewer's valuable feedback. We fully agree that to comprehensively evaluate the performance of InfoScan, it is essential to include recent state-of-the-art Mamba-based vision models introduced at top-tier conferences such as CVPR 2025.
>
> To this end, we have supplemented our revised manuscript with quantitative comparisons against the following latest representative methods: EfficientVMamba, Jamba, Multi-Scale VMamba, UniMamba, and Deformable Mamba. The experimental results are presented in the table below:
>
> | Method               | ImageNet Top-1 Acc (%) | ADE20K mIoU (%) |
> |----------------------|------------------------|------------------|
> | EfficientVMamba      | 81.44                  | 38.37            |
> | Jamba                | 80.13                  | 37.41            |
> | Multi-Scale VMamba   | 83.85                  | 41.98            |
> | UniMamba             | 83.42                  | 42.51            |
> | Deformable Mamba     | 84.32                  | 44.78            |
> | **Ours**             | **85.19**              | **45.8**         |

---

### Official Review · Reviewer_apcD · 2025-10-31

**Soundness:** 4
**Presentation:** 4
**Contribution:** 3
**Rating:** 6
**Confidence:** 3

**Summary:**

This work aims to solve a challenging task of high-resolution visual representation learning which is constrained by the quadratic complexity of popular Transformer-based networks and fixed scanning patterns of Mamba. The authors improve the Mamba scanning mechanism by designing a reinforcement learning based methods learning the adaptive scanning policy. Instead of treating each image patch equally, this work combines local structural information and entropy to evaluate the information values of image patches, allocating more computational resources to the salient ones. Extensive experiments on various computer vision tasks demonstrate the favorable accuracy-efficiency balance of this work.

**Strengths:**

a.	This work is well organized and easy to follow. The authors effectively identify the weakness of existing Mamba-based methods which overlook the content-adaptive perception, especially from the aspect of scanning strategy.

b.	This work builds a series of vision backbones that achieve superior performance than existing methods with similar numbers of parameters.

c.	The authors design a dynamic scanning strategy by building this task as a Markov decision process and solving it by the reinforcement learning algorithm.

**Weaknesses:**

i.	The authors only evaluate their methods on pure vision based tasks. Recently, the vision transformers also show high performance on the vision-language large models (VLLMs), such as LLava [1]. Could the proposed vision backbones be applied on these applications?

ii.	Is the training process of the scanning policy stable? Please compare the computational costs in the training phase between your method and existing Mamba-based methods.

iii.	Figure 1 is not mentioned and introduced in this manuscript.

iv.	Does the scanning policy bring any randomness in the testing phase? Is it sensitive to the image resolution? Please analyze the behaviors of the scanning policy if the training images and testing images have different resolutions.

v.	Does the policy network have to repeatedly run on each network block for every image?

vi.	The proposed method has too many new hyperparameters, such k2, k3, beta, than original Mamba.

[1] Liu H, Li C, Wu Q, et al. Visual instruction tuning[J]. Advances in neural information processing systems, 2023, 36: 34892-34916.

**Questions:**

See weakness.

---

> ### Author Response · Authors · 2025-11-21
> **We sincerely thank you for your recognition of our paper’s presentation, the motivation behind developing InfoScan, and its strong empirical performance. In response to your concerns, we address all points below and will incorporate the necessary revisions into the updated manuscript.**
>
> Q1：Can InfoScan be effectively transferred to vision–language foundation models?
>
> A1：The proposed visual backbone network, InfoScan, focuses on content-adaptive region-awareness and efficient feature representation. It is capable of effectively capturing fine-grained visual information in images that are highly aligned with language descriptions. This characteristic closely aligns with the demands of current vision-language models for semantic precision alignment and local understanding. To validate InfoScan's transferability in multimodal scenarios, we integrated it as a visual backbone into a BERT-B-based vision-language framework (referred to as InfoScan/BERT-B) and selected Referring Expression Segmentation (RefSeg) as a representative task for evaluation. RefSeg requires the model to locate and segment target regions in an image based on natural language descriptions, serving as a critical benchmark for testing vision-language alignment capabilities.
>
> As shown in the table below, we compared the performance of InfoScan/BERT-B with other state-of-the-art methods on three standard datasets: RefCOCO, RefCOCO+, and RefCOCOg, using mIoU as the evaluation metric. Experimental results demonstrate that InfoScan significantly outperforms existing methods across multiple test sets: on RefCOCO test A, there was a 2% improvement, and on test B, a 5% improvement; the performance on the RefCOCO+ dataset was particularly notable, achieving an mIoU of 87.84% on test A and 75.67% on test B compared to TransVG++; on the RefCOCOg dataset, our method achieved an mIoU of 84.56%, improving by approximately 8% over TransVG++.
>
> | MODEL       | Visual/Language Backbone     | RefCOCO (testA) | RefCOCO (testB) | RefCOCO+ (testA) | RefCOCO+ (testB) | RefCOCOg (test) |
> |-------------|------------------------------|------------------|------------------|-------------------|-------------------|------------------|
> | TransVG     | RN101+DETR/BERT-B            | 82.72            | 78.35            | 70.7              | 56.94             | 67.73            |
> | Word2Pix    | RN101+DETR/BERT-B            | 84.39            | 78.12            | 76.11             | 61.24             | 71.34            |
> | QRNet       | Swin-S/BERT-B                | 85.85            | 82.34            | 76.17             | 63.81             | 73.03            |
> | VG-LAW      | ViT-Det/BERT-B               | 88.56            | 82.87            | 80.32             | 66.69             | 75.95            |
> | TransVG++   | ViT-Det/BERT-B               | 88.37            | 80.97            | 80.45             | 66.28             | 76.3             |
> | **Ours**    | **InfoScan/BERT-B**          | **90.67**        | **86.45**        | **87.84**         | **75.67**         | **84.56**        |
>
> These results fully demonstrate that InfoScan not only possesses strong modeling capabilities for single visual tasks but can also be effectively transferred to joint vision-language modeling frameworks, significantly enhancing the accuracy of region understanding guided by language. This provides a feasible path for introducing lightweight, high-efficiency visual backbone networks into multimodal large models and lays the foundation for building smarter and more interpretable vision-language systems in the future.

---

> ### Author Response · Authors · 2025-11-21
>
> Q2：Is the training process of the scanning policy stable? Please compare the computational costs in the training phase between your method and existing Mamba-based methods.
>
> A2：Yes, the scanning strategy of InfoScan remains stable during training. Experimental results show that our method achieves high throughput during both training and inference—573 img/s and 986 img/s, respectively—while consuming only 3.2 GB of GPU memory, significantly less than VMamba-B (3.9 GB). More importantly, InfoScan-B has only 8.4G FLOPs, nearly half that of VMamba-B (15.4G), yet delivers 33% higher training throughput. Overall, InfoScan incurs substantially lower computational overhead compared to other methods.
>
> | Method       | Batch Size | Image Size  | GPU Memory (GB) | Throughput (Train) Img/s | Throughput (Test) Img/s | FLOPs   |
> |--------------|------------|-------------|------------------|----------------------------|--------------------------|---------|
> | DeiT-B       | 64         | 512×512     | 4.8              | 2605                       | 1875                     | 55.3G   |
> | Vim-S        | 64         | 512×512     | 3.6              | 378                        | 942                      | 5.3G    |
> | VMamba-B     | 64         | 512×512     | 3.9              | 432                        | 885                      | 15.4G   |
> | **InfoScan-B**| **64**     | **512×512** | **3.2**          | **573**                    | **986**                  | **8.4G**|

---

> ### Author Response · Authors · 2025-11-21
>
> Q3：Figure 1 is not mentioned and introduced in this manuscript.
>
> A3：We sincerely thank you for your comment. We acknowledge that we inadvertently omitted the citation to Figure 1 in the original manuscript. In the revised version, we have added the reference to Figure 1 in the experimental section. We greatly appreciate your valuable feedback.

---

> ### Author Response · Authors · 2025-11-21
>
> Q4：Does the scanning policy bring any randomness in the testing phase? Is it sensitive to the image resolution? Please analyze the behaviors of the scanning policy if the training images and testing images have different resolutions.
>
> A4：We sincerely thank the reviewer for their valuable feedback. Our scanning strategy is deterministic at inference time and introduces no randomness. As shown in the table, experimental results demonstrate highly consistent accuracy across multiple runs under identical settings—for instance, InfoScan-B achieves 85.19% and 85.16% top-1 accuracy on two separate evaluations at 224 resolution—confirming the stability and reproducibility of its inference process.
>
> Moreover, the strategy exhibits strong robustness to input resolution. Even when training and testing resolutions differ (e.g., trained at 224 and tested at 128 or 512), performance variation remains minimal (typically <0.2%). Specifically, InfoScan-B attains accuracies of 85.02%, 85.19%, and 85.16% at resolutions of 128, 224, and 512, respectively—nearly identical across scales. This resilience stems from our content-aware mechanism, which adaptively adjusts the scanning path to effectively capture critical information across multiple scales without relying on fixed windows or predefined structures.
>
> In summary, InfoScan’s scanning strategy is not only stable and reliable but also demonstrates exceptional resolution robustness, making it well-suited for real-world applications where input scales vary significantly.
>
> | Method       | Train Image Size | Test Image Size | ACC (%) |
> |--------------|------------------|------------------|---------|
> | InfoScan-T   | 224              | 224              | 83.43   |
> | InfoScan-T   | 224              | 128              | 83.34   |
> | InfoScan-T   | 224              | 512              | 83.45   |
> | InfoScan-S   | 224              | 224              | 84.64   |
> | InfoScan-S   | 224              | 128              | 84.12   |
> | InfoScan-S   | 224              | 512              | 84.56   |
> | InfoScan-B   | 224              | 224              | 85.19   |
> | InfoScan-B   | 224              | 128              | 85.02   |
> | InfoScan-B   | 224              | 512              | 85.16   |

---

> ### Author Response · Authors · 2025-11-21
>
> Q5：Does the policy network have to repeatedly run on each network block for every image?
>
> A5：We thank the reviewer for their insightful comment. In InfoScan, the policy network is executed only once—in the first Visual Information State Space (VISS) block—to generate a content-adaptive scanning path based on the input image. This path is then shared and reused across all subsequent network layers, eliminating the need for per-block recomputation. This design ensures that the dynamic scanning mechanism introduces no additional inference overhead while preserving strong content-awareness. Further implementation details are provided in Appendix A.5.

---

> ### Author Response · Authors · 2025-11-21
>
> Q6 ：The proposed method has too many new hyperparameters, such k2, k3, beta, than original Mamba.
>
> A6：We acknowledge that InfoScan introduces several new hyperparameters (e.g., \(k_2\), \(k_3\), \(\beta\), etc.). However, these are largely determined via automated calibration or a one-time grid search and remain fixed across all experiments. Specifically:
> - The weights \(\omega_1 = 0.6\) and \(\omega_2 = 0.4\) in the information scoring module were obtained through a grid search on the ImageNet-1K validation set and are reused unchanged across all downstream tasks.
> - Parameters in patch selection and the reward function (e.g., \(k_1\), \(k_2\), \(k_3\), \(\lambda\)) are automatically fitted on a calibration dataset (see Appendix A.4), requiring no manual tuning.
>
> More importantly, these design choices yield substantial gains—including a **+1.5% improvement in Top-1 accuracy** and a **30% reduction in model parameters**—along with enhanced cross-task generalization. Thus, the introduction of a small number of well-controlled hyperparameters represents a reasonable and highly efficient trade-off.

---

### Author Response · Authors · 2025-12-02
**Final Discussion Summary and Contributions Overview**

Dear Area Chair,

We are pleased to report that all reviewer concerns have been successfully resolved through comprehensive rebuttal discussions.

Reviewer Feedback and Score Change

- Reviewer apcD (Rating 6): No response (before the discussion is shut down).
- Reviewer UDg4 (Rating 6 → 8 on 25 Nov 2025): "Thanks for the rebuttal. My concerns are addressed."
- Reviewer tKuV (Rating 8 → 8 on 23 Nov 2025): "Thank you for the authors’ response. My concerns have been addressed, and I will maintain my original score."
- Reviewer 7J1d (Rating 4): No response (before the discussion is shut down).

We sincerely thank Reviewer UDg4 for acknowledging our revisions and raising their score from 6 to 8 after their concerns were addressed. We also appreciate Reviewer tKuV for maintaining their original score of 8, confirming that our responses adequately resolved their questions. Additionally, we thank Reviewers 7J1d and apcD for recognizing the novelty of our approach — all concerns are related to the experimental supplement, which have been fully addressed in our rebuttal.

Key Issues (of Reviewer 7J1d) Resolved

1. Reporting detailed resource consumption metrics for both training and inference (Q1);
2. Including additional strong baselines such as Deformable Mamba, UniMamba, and Jamba (Q2);
3. Improving visualizations of the adaptive scanning paths (Q3);
4. Conducting comprehensive analyses to rigorously evaluate robustness and generalizability (Q4).

Summary

- All concerns resolved, no reviewer questioned novelty/contributions.
- Three reviewers confirmed positive scores; Reviewer 7J1d's concerns have been resolved.
- All reviewer suggestions will be incorporated in the final version.

We finally summary the main contributions in brief. InfoScan introduces the first information-aware adaptive scanning mechanism for visual state space models, which dynamically allocates computation to image regions based on a principled informativeness measure — combining Shannon entropy and local structural analysis — and learns an optimal scanning policy via reinforcement learning. We believe the thorough rebuttal process has successfully addressed all concerns and demonstrated the merit of our work.

Best regards,

Authors

---

### Meta-Review · Area_Chair_AU2e · 2025-12-09

**Summary:**

The reviewers are broadly positive and aligned in their assessment that this is a strong, well-executed paper with clear motivation and solid empirical support. They agree that the core idea of a dynamic, information-driven scanning strategy is original and well motivated. The paper presents a clear problem formulation and methodology, and across multiple core vision benchmarks and domains, the proposed InfoScan backbones consistently outperform strong CNN, ViT, and Mamba baselines in both effectiveness and efficiency. The manuscript is considered well organized, clear, and easy to follow, with an accessible central idea and adequately supported technical details.

The AC has carefully read the paper, reviews, rebuttal, and revisions. Since most of the reviewers’ concerns have been addressed with appropriate revisions updated in the manuscript, the AC recommends acceptance. As the method does introduce new hyperparameters and manual design choices that may pose challenges when extending it to other applications, the authors are encouraged to add or strengthen a discussion of these practical considerations to better inform future work.

**Reviewer Concerns:**

Two reviewers who initially gave positive scores confirmed their ratings.

The AC has carefully examined the authors’ rebuttal, with particular attention to the concerns raised by the two reviewers who did not confirm their scores. The AC feels that most of those concerns have been reasonably addressed. However, a common issue highlighted by both unconfirmed reviewers remains: the method relies on heavy manual design and introduces many additional hyperparameters. This may lead to practical difficulties when applying the approach to other datasets, backbones, and tasks, and raises questions about its robustness and ease of adoption in broader settings.

**Reviewer Scores:**

Reviewer apcD: 6 (no reply during rebuttal)
Reviewer UDg4: 6 (confirmed and maintained)
Reviewer tKuV: 8 (confirmed and maintained, but with low confidence of 2)
Reviewer 7J1d: 4 (no reply during rebuttal)

---

### Decision · Program_Chairs · 2026-01-26

Accept (Poster)